# Online Agnostic Boosting via Regret Minimization

**Nataly Brukhim**
Google AI Princeton
Princeton University
Department of Computer Science
nbrukhim@princeton.edu

**Xinyi Chen**
Google AI Princeton
Princeton University
Department of Computer Science
xinyic@princeton.edu

**Elad Hazan**
Google AI Princeton
Princeton University
Department of Computer Science
ehazan@princeton.edu

**Shay Moran**[*]
Department of Mathematics
Technion - Israel Institute of Technology
smoran@technion.ac.il

## Abstract

Boosting is a widely used machine learning approach based on the idea of aggregating weak learning rules. While in statistical learning numerous boosting methods exist both in the realizable and agnostic settings, in online learning they exist only in the realizable case. In this work we provide the first agnostic online boosting algorithm; that is, given a weak learner with only marginally-better-than-trivial regret guarantees, our algorithm boosts it to a strong learner with sublinear regret.

Our algorithm is based on an abstract (and simple) reduction to online convex optimization, which efficiently converts an arbitrary online convex optimizer to a boosting algorithm. Moreover, this reduction extends to the statistical as well as the online realizable settings, thus unifying the 4 cases of statistical/online and agnostic/realizable boosting.

## 1 Introduction

Boosting is a fundamental methodology in machine learning that can boost the accuracy of weak learning rules into a strong one. Boosting was first studied in the context of (realizable) PAC learning in a line of seminal works which include the celebrated Adaboost algorithm, as well an many other algorithms with various applications (see e.g. [29, 33, 17, 19]). It was later adapted to the agnostic PAC setting and was extensively studied in this context as well [7, 31, 21, 27, 30, 26, 28, 16, 13, 18]. More recently, [14] and [9] studied boosting in the context of online prediction and derived boosting algorithms in the realizable setting (a.k.a. mistake-bound model).

In this work we study agnostic boosting in the online setting: let $\mathcal{H}$ be a class of experts and assume we have oracle access to a weak online learner for $\mathcal{H}$ with a non-trivial (yet far from desired) regret guarantee. The goal is to convert it into a strong online learner for $\mathcal{H}$ that exhibits vanishing regret.

**Why online agnostic boosting?** The setting of online realizable boosting poses a restriction on the possible input sequences: there must be an expert that attains near-zero mistake-bound on the input sequence. This is a non-standard assumption in online learning. In contrast, in the online agnostic setting we consider, there is *no restriction on the input sequence and it can be chosen adversarially*.

---

[*]The research was done while author was co-affiliated with Google AI Princeton.

**Applications of online agnostic boosting.** Apart from being a fundamental question in a well-studied learning setting, let us mention a few concrete incentives to study online agnostic boosting:

- **Differential privacy and online learning:** A recent line of work revealed deep connections between online learning and differentially private learning [5, 1, 6, 10, 32, 25, 22, 11]. In fact, these two notions are equivalent in the sense that a class $\mathcal{H}$ can be PAC learned by a differentially private algorithm if and only if it can be learned in the online setting with vanishing regret [6, 11]. However, the above equivalence is only known to hold from an information theoretic perspective, and deriving efficient reductions between online and private learning is an open problem [32]. The only case where an efficient reduction is known to exist is in converting a *pure private learner* to an online learner in the realizable setting [22]. This reduction relies heavily on the realizable online boosting algorithm by [9]. Moreover, the derivation of an agnostic online boosting algorithm is posed by [22] as an open problem towards extending their reduction to the agnostic setting.

- **Time series prediction and online control:** Recent machine learning literature considered the problem of controlling a dynamical system from the lens of online learning and regret minimization, see e.g. [3, 4, 24] and referenced work therein. The online learning approach also gave rise to the first boosting methods in this context [2], and demonstrates the potential impact of boosting in the online setting. Thus, the current work aims at continuing the development of the boosting methodology in online learning, starting from the basic setting of learning from expert advice.

## 1.1 Main results

**The weak learning assumption.** In this paper we follow the same formulation as [28] used for boosting in the agnostic statistical setting. Towards this end, it is convenient to measure the performance of online learners using *gain* rather than loss: let $(x_1, y_1) \ldots (x_T, y_T) \in \mathcal{X} \times \{\pm 1\}$ be an (adversarial and adaptive) input sequence of examples presented to an online learning algorithm $\mathcal{A}$; that is, in each iteration $t = 1 \ldots T$, the adversary picks an example $(x_t, y_t)$, then the learner $\mathcal{A}$ first gets to observe $x_t$, and predicts (possibly in a randomized fashion) $\hat{y}_t \in \{\pm 1\}$, and lastly it observes $y_t$ and gains a reward of $y_t \cdot \hat{y}_t$. The goal of the learner is to maximize the total gain (or correlation), given by $\sum_t y_t \cdot \hat{y}_t$. Note that this is equivalent to the often-used notion of *loss* where in each iteration the learner suffers a loss of $1[y_t \neq \hat{y}_t]$ and its goal is to minimize the accumulated loss $\sum_t 1[y_t \neq \hat{y}_t]$. [2]

**Definition 1** (Agnostic Weak Online Learning). *Let $\mathcal{H} \subseteq \{\pm 1\}^{\mathcal{X}}$ be a class of experts, let $T$ denote the horizon length, and let $\gamma > 0$ denote the advantage. An online learning algorithm $\mathcal{W}$ is a $(\gamma, T)$-agnostic weak online learner (AWOL) for $\mathcal{H}$ if for any sequence $(x_1, y_1), ..., (x_T, y_T) \in \mathcal{X} \times \{\pm 1\}$, at every iteration $t \in [T]$, the algorithm outputs $\mathcal{W}(x_t) \in \{\pm 1\}$ such that,*

$$\mathbb{E}\left[\sum_{t=1}^T \mathcal{W}(x_t)y_t\right] \geq \gamma \max_{h \in \mathcal{H}} \mathbb{E}\left[\sum_{t=1}^T h(x_t)y_t\right] - R_{\mathcal{W}}(T),$$

*where the expectation is taken w.r.t the randomness of the learner $\mathcal{W}$ and that of the possibly adaptive adversary, and $R_{\mathcal{W}} : \mathbb{N} \to \mathbb{R}_+$ is the additive regret: a non-decreasing, sublinear function of $T$.*

Note the slight abuse of notation in the last definition: an online learner $\mathcal{W}$ is not an "$\mathcal{X} \to \{\pm 1\}$" function; rather it is an algorithm with an internal state that is updated as it is fed new examples. Thus, the prediction $\mathcal{W}(x_t)$ depends on the internal state of $\mathcal{W}$, and for notational convenience we avoid reference to the internal state.

Our agnostic online boosting algorithm has oracle access to $N$ weak learners and predicts each label by combining their predictions. The number of weak learners $N$ is a meta-parameter which can be tuned by the user according to the following trade-off: on the one hand, the regret bound improves as $N$ increases, and on the other hand, a larger number of weak learners is more costly in terms of computational resources.

**Theorem 2** (Agnostic Online Boosting)**.** *Let $\mathcal{H}$ be a class of experts, let $T \in \mathbb{N}$ denote the horizon length, and let $\mathcal{W}_1, \dots, \mathcal{W}_N$ be $(\gamma, T)$-**AWOL** for $\mathcal{H}$ with advantage $\gamma$ and regret $R_{\mathcal{W}}(T) = o(T)$ (see Definition 1). Then, there exists an online learning algorithm, which has oracle access to each $\mathcal{W}_i$, and has expected regret of at most*

$$\frac{R_{\mathcal{W}}(T)}{\gamma} + O\Big(\frac{T}{\gamma\sqrt{N}}\Big).$$

To exemplify the interplay between $R_{\mathcal{W}}(\cdot)$ and $N$, imagine a scenario where $R_{\mathcal{W}}(T) \approx \sqrt{T}$ (as is often the case for regret bounds). Then, setting $N \approx T$ gives that the overall regret remains $\approx \sqrt{T}$. By setting both $T$ and $N$ to be $O(\frac{1}{\gamma^2 \epsilon^2})$ for any $\epsilon > 0$, an average regret of $\epsilon$ is obtained.

**An abstract framework for boosting.** Boosting and Regret Minimization algorithms are intimately related. This tight connection is exhibited in statistical boosting (see [20, 19, 34]). For example, AdaBoost can be interpreted as applying the Hedge algorithm to iteratively update probability weights associated with the training examples [18]. Our algorithm is inspired by this fruitful connection and utilizes it. We derive a general framework which reduces boosting to online convex optimization. Moreover, this reduction applies to 4 learning settings: realizable-statistical, realizable-online, agnostic-statistical, and agnostic-online.

We note that in the agnostic boosting settings, both the assumption and the goal are stronger compared to the realizable case; an agnostic weak learner is assumed (which is stronger than a realizable weak learner), but the aim is to learn arbitrary distributions (which is a more challenging task than only learning realizable distributions). Therefore, a boosting algorithm for the realizable setting does not trivially follow from an agnostic boosting algorithm. A similar argument holds for the statistical vs. online boosting settings.

The general framework we derive in this work does apply to all the 4 aforementioned learning settings, with only slight modification to the algorithm's update rule. On a high-level, these modification are based on the following observations:

1. Agnostic/Realizable: in the realizable setting the weights of instances (i.e. the $x$'s) are updated (e.g., in Adaboost [34]). In the agnostic setting the weights of labels (i.e. the $y$'s) are typically updated instead [16, 28]. Therefore, instances and labels are exchanged.

2. Statistical/Online: in the statistical setting, in each iteration a weak-hypothesis is produced and the weights of all examples are updated. In the online setting in each iteration a new example is processed and the weights corresponding to all weak-learners are updated [9]. Therefore, examples and weak learners are exchanged.

Thus, there is an interesting duality which "converts" between the 4 settings by replacing reweighting (realizable) with relabeling (agnostic), and between updating all learners per example (online) and updating all examples per learner (statistical). Our main contribution is showing that the fashion in which reweighting/relabelling is performed, which was previously given by ad hoc update-rules, can be abstracted to an application of an arbitrary online convex optimization algorithm.

Our general framework result may not come as a surprise given the well known connections between boosting and online convex optimization. However, we stress that the general reduction established here does introduce technical challenges. For instance, the final output of the algorithm is not obtained via a standard weighted majority-vote, but rather a different projected aggregation of the weak learners' predictions. Thus, albeit our unified framework being simple and intuitive, the analysis is not straightforwardly derived.

**Paper structure.** In the next subsection we discuss related work. The main result of our agnostic online boosting algorithm, and the proof of Theorem 2, are given in Section 2. In Section 3 we demonstrate a general reduction, in the statistical setting, from both the agnostic (Subsection 3.1), and realizable (Subsection 3.2) boosting settings, to online convex optimization. Then, in Section 4, we give a similar result for the online realizable boosting setting.

## 1.2 Related work

Several previous works derived agnostic boosting algorithms in the statistical setting [26, 27, 28, 16]. However, previous works on online boosting have focused only on the realizable (mistake-bound) setting [14, 9]. The work by [14] provides a boosting algorithm inspired by classical algorithms in the realizable-statistical setting. Their work was later extended to an optimal and adaptive online boosting algorithm [9]. Although the work by [9] does not explicitly assume realizability, we remark that Equation (1) in [9] amounts to realizability: indeed, it assumes that the weak learner makes at most $(0.5 - \gamma)T + o(T)$ mistakes on every sequence of input examples $(x_1, y_1)...(x_T, y_T)$. This clearly cannot apply in an agnostic setting, since for a fixed $\gamma > 0$, if the input sequence is drawn randomly then with high probability the number of mistakes will concentrate around $0.5T$ and Equation (1) will fail. Thus, in order for Equation (1) to hold, one must restrict the set of allowable sequences, which essentially amounts to realizability with respect to some class. In contrast, our work focuses on the agnostic (regret-bound) setting, where the input sequence is not restricted and can be adversarial.

Another contribution of this work is the above-mentioned general framework which reduces boosting to online convex optimization. We note however that this generality comes with a cost: in particular, in the 3 settings that were previously studied, the boosting algorithms derived from the general framework exhibit, in some aspects, inferior guarantees compared to the state-of-the-art (e.g., Adaboost [34] in the realizable-statistical case, [9] in the realizable-online case, and [28] in the agnostic-statistical case). We compare them in more detail in the following sections. It is therefore presumable that our online agnostic boosting algorithm can also be improved, e.g. by improving the oracle/regret bounds or adding adaptive capabilities. We leave these goals for future work.

Several other works [8, 2] studied online boosting under real-valued loss functions. The main difference from our work is in the weak learning assumption: they consider weak learners that are in fact strong online learners for a base class of regression functions. The boosting procedure produces an online learner for a bigger class which consists of the linear span of the base class. This is different from the setting considered here where the class is fixed, but the regret bound is being boosted.

A main motivation of this work is the connection between boosting and regret minimization. This builds on and is inspired by previous works demonstrating this relationship. We refer the reader to the book by [34] (Chapter 6) for an excellent presentation of this relationship in the context of Adaboost.

## 2 Online agnostic boosting

In this section, we formally present our main result which enables converting an online convex optimizer to an online agnostic booster. We begin by providing an overview of online convex optimization, and then proceed to describe our main algorithm.

**Online Convex Optimization.** (see e.g. [23]). In the Online Convex Optimization (OCO) framework, an online player iteratively makes decisions from a compact convex set $\mathcal{K} \subset \mathbb{R}^d$. At iteration $i = 1, ..., N$, the online player chooses an action $p^i$ in the decision set $\mathcal{K}$, and at the same time the adversary reveals a cost function $\ell^i : \mathcal{K} \to \mathbb{R}$, chosen from a family $\mathcal{F}$ of bounded convex functions over $\mathcal{K}$. We denote an algorithm $\mathcal{A}$ in this setting as a $(\mathcal{K}, N)$-Online Convex Optimizer (OCO), if the regret $R_{\mathcal{A}}(N)$ is a non-decreasing, sublinear function of $N$, where the regret is defined as the excess loss over the best single action in hindsight over the decision set $\mathcal{K}$:

$$R_{\mathcal{A}}(N) = \sum_{i=1}^{N} \ell^i(p^i) - \min_{p \in \mathcal{K}} \sum_{i=1}^{N} \ell^i(p). \tag{1}$$

The OCO framework generalizes the statistical learning framework as it permits the adversary to adapt to the actions of the player. Under this setting, the best action in hindsight is a competitive benchmark, and a sublinear regret guarantees vanishing average excess loss over time.

**Main Algorithm.** Our main algorithm is formally described in Algorithm 1. The booster has black-box oracle access to two types of auxiliary algorithms: $N$ instances of an online weak learning algorithm, denoted $\mathcal{W}_1, \ldots, \mathcal{W}_N$, and an online-convex optimizer, denoted by $\mathcal{A}$. The algorithm proceeds in rounds, where in each round $t$ it observes an example $(x_t, y_t)$ and sequentially updates each weak learner $\mathcal{W}_i$ by feeding it with $(x_t, y_t^i)$, where $y_t^i$ is a re-labeling. The role of the OCO is to

determine the $y_t^i$'s (via the parameters $p_t^i$'s, see Line 4). Intuitively, it guides each weak learner $\mathcal{W}_i$ to correct for mistakes of the preceding learners. We stress that the OCO is restarted in each round.

**Randomized Majority-Vote/Projection.** At the beginning of each iteration, the boosting algorithm aggregates the weak learners' predictions using a randomized projection "$\Pi$". For any $z \in \mathbb{R}$, denote by $\Pi(z)$ the following random label:

$$\Pi(z) = \begin{cases} \text{sign}(z) & \text{if } |z| \geq 1 \\ +1 & \text{w.p. } \frac{1+z}{2} \\ -1 & \text{w.p. } \frac{1-z}{2} \end{cases} \tag{2}$$

---

**Algorithm 1** Online Agnostic Boosting with OCO

1: **for** $t = 1, \ldots, T$ **do**
2:     Get $x_t$, predict: $\hat{y}_t = \Pi\left(\frac{1}{\gamma N} \sum_{i=1}^N \mathcal{W}_i(x_t)\right)$.
3:     **for** $i = 1, \ldots, N$ **do**
4:         If $i > 1$, set $p_t^i = \mathcal{A}(\ell_t^1, \ldots, \ell_t^{i-1})$.   \\Note that $\mathcal{A}$ is restarted at each time step $t$.
5:         Else, set $p_t^1 = 0$
6:         Set next loss: $\ell_t^i(p) = p(\frac{1}{\gamma}\mathcal{W}_i(x_t)y_t - 1)$.
7:         Pass $(x_t, y_t^i)$ to $\mathcal{W}_i$, where $y_t^i$ is a random label s.t. $\mathbb{P}[y_t^i = y_t] = \frac{1+p_t^i}{2}$.
8:     **end for**
9: **end for**

---

*Figure 1:* The algorithm is given oracle access to $N$ instances of a $(\gamma, T)$-AOWL algorithm, $\mathcal{W}_1, \ldots, \mathcal{W}_N$ (see Definition 1), and to a $([-1,1], N)$-OCO algorithm $\mathcal{A}$ (see Equation 1). The prediction "$\Pi(\frac{1}{\gamma N} \sum_{i=1}^N \mathcal{W}_i(x_t))$" in line 2 is a projected majority-vote (see Equation 2).

We now state and prove the regret bound for Algorithm 1.

**Proposition 3** (Regret Bound)**.** *The accumulated gain of Algorithm 1 satisfies:*

$$\frac{1}{T} \mathbb{E}\left[\max_{h^* \in \mathcal{H}} \sum_{t=1}^T h^*(x_t)y_t - \sum_{t=1}^T \hat{y}_t y_t\right] \leq \frac{R_{\mathcal{W}}(T)}{\gamma T} + \frac{R_{\mathcal{A}}(N)}{N},$$

*where $(x_t, y_t)$'s are the observed examples, $\hat{y}_t$'s are the predictions, the expectation is with respect to the algorithm and learners' randomness, and $R_{\mathcal{W}}$, $R_{\mathcal{A}}$ are the AWOL, OCO regret terms, respectively.*

*Proof.* The proof follows by combining upper and lower bounds on the expected sum of losses incurred by the OCO algorithm. The bounds follow directly from the weak learning assumption (lower bound) and the OCO guarantee (upper bound). These bounds involve some simple algebraic manipulations. It is convenient to abstract out some of these calculations into lemmas, which are described later in this section.

Before delving into the analysis, we first clarify several assumptions used below. For simplicity of presentation we assume an oblivious adversary, however, using a standard reduction, our results can be generalized to an adaptive one [3]. Let $(x_1, y_1), \ldots, (x_T, y_T)$ be any sequence of observed examples. Observe that there are several sources of randomness at play; the weak learning algorithm $\mathcal{W}_i$'s internal randomness, the random re-labeling (line 6, Algorithm 1), and the randomized prediction (line 7, Algorithm 1). The analysis below is given in expectation w.r.t. all these random variables.

Note the following fact used in the analysis; for all $i \in [N], t \in [T]$, the random variables $\mathcal{W}_i(x_t)$ and $y_t^i$ are conditionally independent given $p_t^i$ and $y_t$. Since $\mathbb{E}[y_t^i|p_t^i, y_t] = p_t^i \cdot y_t$, using the conditional independence, it follows that $\mathbb{E}[\mathcal{W}_i(x_t)y_t^i] = \mathbb{E}[\mathcal{W}_i(x_t)p_t^i y_t]$ (see Lemma 13 in the Appendix). We can now begin the analysis, starting with lower bounding the expected sum of losses, using the weak learning guarantee,

$$\frac{1}{\gamma}\mathbb{E}\left[\sum_{i=1}^N \sum_{t=1}^T \mathcal{W}_i(x_t) \cdot y_t p_t^i\right] = \frac{1}{\gamma} \sum_{i=1}^N \mathbb{E}\left[\sum_{t=1}^T \mathcal{W}_i(x_t) \cdot y_t p_t^i\right] = \frac{1}{\gamma} \sum_{i=1}^N \mathbb{E}\left[\sum_{t=1}^T \mathcal{W}_i(x_t)y_t^i\right]$$

(See Lemma 13)

$$\geq \frac{1}{\gamma} \sum_{i=1}^{N} \Big( \gamma \max_{h \in \mathcal{H}} \mathbb{E}\Big[\sum_{t=1}^{T} h(x_t)y_t^i\Big] - R_{\mathcal{W}}(T) \Big) \quad \text{(Weak Learning (1))}$$

$$\geq \sum_{i=1}^{N} \Big( \max_{h \in \mathcal{H}} \sum_{t=1}^{T} h(x_t) \cdot \mathbb{E}[y_t^i] - \frac{1}{\gamma} R_{\mathcal{W}}(T) \Big)$$

$$\geq \sum_{i=1}^{N} \sum_{t=1}^{T} h^*(x_t) \cdot \mathbb{E}[y_t p_t^i] - \frac{N}{\gamma} R_{\mathcal{W}}(T)$$

$$= \sum_{i=1}^{N} \sum_{t=1}^{T} \mathbb{E}\big[h^*(x_t) \cdot y_t p_t^i\big] - \frac{N}{\gamma} R_{\mathcal{W}}(T),$$

where $h^*$ is an optimal expert in hindsight for the observed sequence of examples $(x_t, y_t)$'s. Thus, we obtain the lower bound on the expected sum of losses $\sum_t \sum_i \ell_t^i(p_t^i)$ (see Line 6 in Algorithm 1 for the definition of the $\ell_t^i$'s), given by,

$$\mathbb{E}[\sum_{t=1}^{T} \sum_{i=1}^{N} \ell_t^i(p_t^i)] \geq \sum_{i=1}^{N} \sum_{t=1}^{T} \mathbb{E}\big[p_t^i(h^*(x_t)y_t - 1)\big] - \frac{N}{\gamma} R_{\mathcal{W}}(T)$$

$$\geq N \sum_{t=1}^{T} (h^*(x_t)y_t - 1) - \frac{N}{\gamma} R_{\mathcal{W}}(T). \qquad \text{(See Lemma 4 below)}$$

For the upper bound, observe that the OCO regret guarantee implies that for any $t \in [T]$, and any $p_t^* \in [-1, 1]$,

$$\mathbb{E}\Big[\frac{1}{N} \sum_{i=1}^{N} \ell_t^i(p_t^i)\Big] \leq p_t^* \Big( \Big( \frac{1}{\gamma N} \sum_{i=1}^{N} \mathbb{E}\big[\mathcal{W}_i(x_t)\big] \Big) y_t - 1 \Big) + \frac{1}{N} R_{\mathcal{A}}(N),$$

Thus, by setting $p_t^*$ according to Lemma 5 (see below, with $\hat{h}(x) := \frac{1}{\gamma N} \sum_{i=1}^{N} \mathbb{E}\big[\mathcal{W}_i(x)\big]$), and summing over $t \in [T]$, we get,

$$\mathbb{E}\Big[\frac{1}{N} \sum_{t=1}^{T} \sum_{i=1}^{N} \ell_t^i(p_t^i)\Big] \leq \sum_{t=1}^{T} (\mathbb{E}[\hat{y}_t]y_t - 1) + \frac{T}{N} R_{\mathcal{A}}(N).$$

By combining the lower and upper bounds for $\mathbb{E}\big[\frac{1}{NT} \sum_t \sum_i \ell_t^i(p_t^i)\big]$, we get,

$$\frac{1}{T} \sum_{t=1}^{T} \mathbb{E}[\hat{y}_t]y_t \geq \frac{1}{T} \sum_{t=1}^{T} h^*(x_t)y_t - \frac{R_{\mathcal{W}}(T)}{\gamma T} - \frac{R_{\mathcal{A}}(N)}{N}.$$

$\square$

It remains to prove two short Lemmas that are used in the proof of the theorem above, as well as in the more general settings in the following sections.

**Lemma 4.** *For any $p \in [-1, 1]$, an example pair $(x, y)$, and $h : \mathcal{X} \to \{-1, 1\}$, we have:*
$$p(h(x)y - 1) \geq h(x)y - 1.$$

*Proof.* Let $z = h(x)y - 1$. Observe that $z \in \{-2, 0\}$. Thus, since $p \in [-1, 1]$, $pz \geq z$. $\square$

**Lemma 5.** *Given an example pair $(x, y)$, and $\hat{h} : \mathcal{X} \to \mathbb{R}$, there exists $p^* \in \{0, 1\}$, such that,*
$$p^*(\hat{h}(x)y - 1) \leq \hat{y}y - 1,$$

*where $\hat{y}_t = \mathbb{E}[\Pi(\hat{h}(x))]$, with expectation taken only w.r.t. the randomness of $\Pi$ (see Definition (2)).*

*Proof.* If $|\hat{h}(x)| \leq 1$, $\hat{y} = \hat{h}(x)$ and by setting $p^* = 1$, the equality follows. Thus, assume $|\hat{h}(x)| > 1$, and consider the following cases:

- If $\hat{h}(x)y - 1 > 0$, then $\hat{y}y - 1 = 0$. Hence, by setting $p^* = 0$, the equality follows.
- If $\hat{h}(x)y - 1 < 0$, then since $|\hat{h}(x)| > 1$ it must be that $\text{sign}(\hat{h}(x))y = -1$, and $\hat{y}y - 1 = -2$. Since $|\hat{h}(x)| > 1$, we have $\hat{h}(x)y - 1 \leq -2$. Hence, by setting $p^* = 1$ the inequality holds.

$\square$

## 2.1 Proof of Theorem 2

The proof of Theorem 2 is a direct corollary of Proposition 3, by plugging *Online Gradient Descent (OGD)* to be the OCO algorithm $\mathcal{A}$ (e.g., see [23] Chapter 3.1): the OGD regret is $O(GD\sqrt{N})$, where $N$ is the number of iterations, $G$ is an upper bound on the gradient of the losses, and $D$ is the diameter of the set $\mathcal{K} = [-1, 1]$. In our setting, $G \leq \frac{2}{\gamma}$, and $D = 2$. Hence, $R_{\mathcal{A}} = O(\sqrt{N}/\gamma)$, and the overall bound on the regret follows.

## 3 Statistical realizable and agnostic boosting

In this section we give an overview of our results in the statistical setting. The formal definitions and results are stated below, in sections 3.1 ans 3.2. The full analysis is deferred to the Appendix.

The algorithms and analysis in the statistical-setting follow the same structure as in the online setting (see the discussion of the abstract framework for boosting in Section 1). To allow the reader to assess the simplicity of the abstraction, we include the pseudo-code for our algorithms in the statistical setting below in Algorithm 2. Notice that, as in the online setting, the difference between the agnostic- and realizable-case boosting algorithms below boils down to a single line.

Note that a caveat of this unified framework is that the oracle and sample complexity bounds we obtain in the realizable setting are inferior compared to the state-of-the-art bounds (see e.g., [34]); in the agnostic setting our bounds match the state-of-the-art bounds [28, 16], however our algorithm lacks adaptivity (whereas, the algorithms by [28, 16] are adaptive). Albeit not achieving quantitative improvement, the applications given below demonstrate the generality of the proposed framework. Finally, let us remark a curious connection between our boosting algorithms in the statistical setting with repeated zero-sum game-play with access to an approximately-best-response oracle. Given an input strategy of the opponent, such an oracle returns a strategy whose payoff competes with the best response up to a multiplicative factor. Interestingly, our algorithm can be seen as an "improper" policy in this repeated-game setting. We refer the reader to the appendix for the formal details.

---

**Algorithm 2** Boosting with OCO

1: **for** $t = 1, \ldots, T$ **do**
2:     Pass $m_0$ examples to $\mathcal{W}$ drawn from the following distribution:
3:     **Realizable**: Draw $(x_i, y_i)$ w.p. $\propto p_t(i)$.
4:     **Agnostic**: Draw $x_i$ w.p. $\frac{1}{m}$, and re-label according to $y_i p_t(i)$.
5:     Let $h_t$ be the weak hypothesis returned by $\mathcal{W}$.
6:     Set loss: $\ell_t(p) = \sum_{i=1}^{m} p(i)(\frac{1}{\gamma} h_t(x_i)y_i - 1)$.
7:     Update: $p_{t+1} = \mathcal{A}(\ell_1, ..., \ell_t)$.
8: **end for**
9: **return** $\bar{h}(x) = \Pi\left(\frac{1}{\gamma T} \sum_{t=1}^{T} h_t(x)\right)$.

---

*Figure 2:* The booster has oracle access to either a $(\gamma, \epsilon_0, m_0)$-AWL (see Definition 6) or a $(\gamma, m_0)$-WL (see Definition 8), both denoted as $\mathcal{W}$. The optimizer is a $(\gamma, \mathcal{K}, T)$-OCO $\mathcal{A}$ (see Definition 1), $\mathcal{K} = [0, 1]^m$ in the realizable case and $\mathcal{K} = [-1, 1]^m$ in the agnostic case. The final predictor (Line 9) applies a projected majority-vote (see Equation 2).

### 3.1 Statistical agnostic boosting

Let $\mathcal{H} \subseteq \{\pm 1\}^{\mathcal{X}}$ be a hypothesis class, and let $\mathcal{D}$ be any a distribution over $\mathcal{X} \times \{\pm 1\}$. Define the correlation of any $h \in \mathcal{H}$ with respect to $\mathcal{D}$ by $\mathrm{cor}_{\mathcal{D}}(h) = \mathbb{E}_{(x,y)\sim\mathcal{D}}[h(x) \cdot y]$.

**Definition 6** (Empirical Agnostic Weak Learning Assumption). *Let* $\mathbf{x} = (x_1 \ldots x_m) \in \mathcal{X}$ *denote an unlabeled sample. A learning algorithm* $\mathcal{W}$ *is a* $(\gamma, \epsilon_0, m_0)$-**agnostic weak learner (AWL)** *for* $\mathcal{H}$ *with respect to* $\mathbf{x}$ *if for any labels* $\mathbf{y} = (y_1, \ldots, y_m)$,

$$\mathbb{E}_{S'}[\mathrm{cor}_{\mu\times\mathbf{y}}(\mathcal{W}(S'))] \geq \gamma \max_{h^*\in\mathcal{H}} \mathrm{cor}_{\mu\times\mathbf{y}}(h^*) - \epsilon_0,$$

*where* $\mu \times \mathbf{y}$ *is the distribution which uniformly assigns to each example* $(x_i, y_i)$ *probability* $1/m$, *and* $S'$ *is an independent sample of size* $m_0$ *drawn from* $\mu \times \mathbf{y}$.

In accordance with previous works, we focus on the setting where $\gamma$ is a small constant (say $\gamma = 0.1$) and $\varepsilon_0 \approx d/\sqrt{m}$, where $d$ is the VC-dimension of $\mathcal{H}$ (see [28] for a detailed discussion). We stress

however that our results apply for any setting of $\gamma, \epsilon_0 \in [0, 1]$. The above weak learning assumption can be seen as an empirical variant of the assumption in [28], where $\mu$ is replaced with the population distribution over $\mathcal{X}$ and the labels $y_i$'s are replaced with an arbitrary classifier $c : X \to \{\pm 1\}$. Both of these assumptions are weaker than the standard agnostic weak learning assumption, for which the guarantee holds with respect to every distribution $\mathcal{D}$ over $\mathcal{X} \times \{\pm 1\}$. It will be interesting to investigate the relationship between the assumption of [28] and our empirical variant, however this is beyond the scope of this work. We now state the empirical error bound for Algorithm 2.

**Theorem 7** (Empirical Agnostic Boosting). *The correlation of $\bar{h}$, output of Algorithm 2, satisfies:*

$$\mathbb{E}\big[\operatorname{cor}_S(\bar{h})\big] \geq \max_{h^* \in \mathcal{H}} \mathbb{E}\big[\operatorname{cor}_S(h^*)\big] - \left(\frac{\epsilon_0}{\gamma} + O\left(\frac{1}{\gamma\sqrt{T}}\right)\right). \tag{3}$$

Observe that by setting $T = O(\frac{1}{\gamma^2 \epsilon^2})$ for any $\epsilon > 0$, an error of $\epsilon$ is obtained. Note that this in fact matches bound on $T$ in the state-of-the-art agnostic boosting method [28]. However, we remark that the method given in [28] is adaptive.

## 3.2 Statistical realizable boosting

**Definition 8** (Empirical Weak Learning Assumption [34]). *Let $\mathcal{H} \subseteq \{\pm 1\}^{\mathcal{X}}$ be a hypothesis class, and let $S = \{(x_1, y_1), \ldots, (x_m, y_m)\} \in \mathcal{X} \times \{\pm 1\}$ be a sample. A learning algorithm $\mathcal{W}$ is a $(\gamma, m_0)$-**weak learner (WL)** for $\mathcal{H}$ with respect to $S$ if for any distribution $\mathbf{p} = (p_1, \ldots, p_m)$ which assigns each example $(x_i, y_i)$ with probability $p_i$,*

$$\mathbb{E}_{S'}\big[\operatorname{cor}_{\mathbf{p}}(\mathcal{W}(S'))\big] \geq \gamma,$$

*where $S'$ is an independent sample of size $m_0$ drawn from $\mathbf{p}$.*

**Theorem 9** (Empirical Realizable Boosting). *The correlation of $\bar{h}$, output of Algorithm 2, satisfies:*

$$\mathbb{E}[\operatorname{cor}_S(\bar{h})] \geq 1 - O\left(\frac{1}{\gamma\sqrt{T}}\right).$$

Observe that by setting $T = O(\frac{1}{\gamma^2 \epsilon^2})$ for any $\epsilon > 0$, at most $\epsilon$ error is obtained. We remark that classical boosting results are able to achieve $\epsilon$ error, by setting $T = O\left(\frac{1}{\gamma^2}\log(\frac{1}{\epsilon})\right)$ [34].

## 3.3 Generalization

Theorems 7 and 9 imply that the correlation of the output hypothesis is competitive with the best hypothesis in $\mathcal{H}$ with respect to the empirical distribution. A similar guarantee with respect to the population distribution can be obtained using a standard sample compression argument (see, e.g. [34, 15]): indeed, the final hypothesis $\bar{h}$ is obtained by aggregating the $T$ weak hypotheses $h_t$'s, each of which is determined by the $m_0$ examples fed to the weak learner. Thus, $\bar{h}$ can be encoded by $T \cdot m_0$ input examples and hence the entire algorithm forms a sample compression scheme of this size. Consequently, setting the input sample size $m = \tilde{O}(T \cdot m_0 / \varepsilon^2)$ yields the same guarantees of Theorems 7, 9 up to an additive error of $\varepsilon$.

# 4 Online realizable boosting

In this section, we give an online realizable boosting algorithm, and state its regret bound. The result follows along similar lines as our main result given in Section 2. We first state the weak learning assumption for the online realizable setting.

**Definition 10** (Online Weak Learning). *Let $\mathcal{H} \subseteq \{\pm 1\}^{\mathcal{X}}$ be a class of experts, let $T$ denote the horizon length, and let $\gamma > 0$ denote the advantage. An online learning algorithm $\mathcal{W}$ is a $(\gamma, T)$-**weak online learner (WOL)** for $\mathcal{H}$ if for any sequence $(x_1, y_1), \ldots, (x_T, y_T) \in \mathcal{X} \times \{\pm 1\}$ that is realizable by $\mathcal{H}$, at every iteration $t \in [T]$, the algorithm outputs $\mathcal{W}(x_t) \in \{\pm 1\}$ such that,*

$$\mathbb{E}\left[\sum_{t=1}^{T} \mathcal{W}(x_t) y_t\right] \geq \gamma T - R_{\mathcal{W}}(T),$$

*where the expectation is taken over the randomness of the weak learner $\mathcal{W}$ and $R_{\mathcal{W}} : \mathbb{N} \to \mathbb{R}_+$ is the additive regret: a non-decreasing, sub-linear function of $T$.*

Recall that the restriction of the sequence $\{(x_t, y_t)\}_{t=1}^T$ to being realizable by $\mathcal{H}$ corresponds to: $\max_{h \in \mathcal{H}} \frac{1}{T} \sum_{t=1}^T h(x_t)y_t = 1$. Similar to the online agnostic case, the booster maintains $N$ instances $\mathcal{W}_1, \dots, \mathcal{W}_N$ of an online weak learning algorithm, and a single instance of the OCO algorithm $\mathcal{A}$. However, in the previous section, the OCO's decision set is $[-1, 1]$, corresponding to its role of relabeling examples in $\pm 1$. In this setting, the decision set of the OCO is $[0, 1]$, corresponding to its role of choosing probability weights for each example. Indeed, observe that Algorithm 1 and Algorithm 3 differ on Line 7.

---

**Algorithm 3** Online boosting with OCO

1: **for** $t = 1, \dots, T$ **do**
2:     Get $x_t$, predict: $\hat{y}_t = \Pi\left(\frac{1}{\gamma N} \sum_{i=1}^N \mathcal{W}_i(x_t)\right)$.
3:     **for** $i = 1, \dots, N$ **do**
4:         If $i > 1$, set $p_t^i = \mathcal{A}(\ell_t^1, \dots, \ell_t^{i-1})$.   \\Note that $\mathcal{A}$ is restarted at each time step $t$.
5:         Else, set $p_t^1 = 1/2$.
6:         Set next loss: $\ell_t^i(p) = p(\frac{1}{\gamma}\mathcal{W}_i(x_t)y_t - 1)$.
7:         Pass $(x_t, y_t)$ to $\mathcal{W}_i$ w.p. $p_t^i$.
8:     **end for**
9: **end for**

---

*Figure 3:* The algorithm is given oracle access to $N$ instances of a $(\gamma, T)$-WOL algorithm, $\mathcal{W}_1, \dots, \mathcal{W}_N$ (see Definition 10), and to a $([0, 1], N)$-OCO algorithm $\mathcal{A}$ (see Equation 1). The prediction "$\Pi\left(\frac{1}{\gamma N} \sum_{i=1}^N \mathcal{W}_i(x_t)\right)$" in line 2 is a projected majority-vote (see Equation 2).

The following theorem formally states the guarantees of the online realizable boosting algorithm:

**Theorem 11.** *The accumulated gain of Algorithm 3 satisfies:*

$$\mathbb{E}\left[\frac{1}{T}\sum_{t=1}^T \hat{y}_t y_t\right] \geq 1 - \left(\frac{\tilde{R}_{\mathcal{W}}(T)}{\gamma T} + \frac{R_{\mathcal{A}}(N)}{N}\right).$$

*where $(x_t, y_t)$'s are the observed examples, $\hat{y}_t$'s are the predictions, the expectation is with respect to the algorithm and weak learners' randomness, $\tilde{R}_{\mathcal{W}}(T) := 2R_{\mathcal{W}}(T) + \tilde{O}(\sqrt{T})$, and $R_{\mathcal{W}}$ and $R_{\mathcal{A}}$ are the regret terms of the weak learner and the OCO, respectively.*

The proof follows similarly to that of Proposition 3, and is deferred to the Appendix. By plugging *Online Gradient Descent (OGD)* to be the OCO algorithm $\mathcal{A}$, we get that $R_{\mathcal{A}}(N) = O(\frac{\sqrt{N}}{\gamma})$. Thus, by Theorem 11, we obtain that the overall error is $O(\frac{1}{\gamma\sqrt{T}} + \frac{1}{\gamma\sqrt{N}})$. Note that by setting $N = O(\frac{1}{\gamma^2 \epsilon^2})$ and $T = O(\frac{1}{\gamma^2 \epsilon^2})$, the mistake-bound is at most $\varepsilon \cdot T$. This is worse by a factor of $1/\epsilon^2$ than the bound given by [9], yielding $T = \tilde{O}(\frac{1}{\gamma^2 \epsilon})$ and $N = O(\frac{1}{\gamma^2}\ln(\frac{1}{\epsilon}))$, which is optimal.

## 5 Discussion

We have presented the first boosting algorithm for agnostic online learning. In contrast to the realizable setting, we do not place any restrictions on the online sequence of examples. It remains open to prove lower bounds on online agnostic boosting as a function of the natural parameters of the problem and/or improve our upper bounds.

## Broader Impact

There are no foreseen ethical or societal consequences for the research presented herein.

## Acknowledgments and Disclosure of Funding

NB, XC, and EH acknowledge the support of NSF grant 1704860 and Google. In addition, XC acknowledges the support of the NSF Graduate Research Fellowships Program. This work was done when SM was a visiting scholar at Google.

## Footnotes

[2] Indeed, $y_t\hat{y}_t = 1 - 2 \cdot 1[y_t \neq \hat{y}_t]$ since $y_t, \hat{y}_t \in \{\pm 1\}$. Therefore, the accumulated loss and correlation are affinely related by $\sum y_t \cdot \hat{y}_t = T - 2 \cdot \sum_t 1[y_t \neq \hat{y}_t]$.

[3]See discussion in [12], Pg. 69, as well as Exercise 4.1 formulating the reduction.

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
