[Supplementary Material]

# A   Statistical Boosting via Improper Game Playing

In this section we first give a game-theoretic perspective of our method when applied to the statistical setting (Subsection A.1). We then demonstrate a general reduction from both the agnostic (Subsection 3.1), and realizable (Subsection 3.2) boosting settings, to online convex optimization. The following algorithm is given as input a sample $S = (x_1, y_1), \ldots, (x_m, y_m) \in \mathcal{X} \times \mathcal{Y}$, and has a black-box access to two auxiliary algorithms: a weak learner, and an online-convex optimizer. Note that this in fact defines a family of boosting algorithms, depending on the choice of the online-convex optimizer.

The algorithm iteratively chooses a weak hypothesis by applying re-weighting / re-labeling examples, in the realizable / agnostic setting, respectively. That is, the OCO algorithm $\mathcal{A}$ is (only) used to determine the $p_t$ parameters which correspond to either a re-weighting or re-labeling of the $m$ training examples, at iteration $t$. Intuitively, each weak hypothesis $h_t$ was chosen to correct for mistakes of the previously chosen hypotheses. A formal description is provided in Algorithm 4.

---

**Algorithm 4** Boosting with OCO

---

1: **for** $t = 1, \ldots, T$ **do**
2:     Pass $m_0$ examples to $\mathcal{W}$ drawn from the following distribution:
3:     **Realizable**: Draw $(x_i, y_i)$ w.p. $\propto p_t(i)$[4].
4:     **Agnostic**: Draw $x_i$ w.p. $\frac{1}{m}$, and re-label according to $y_i p_t(i)$.
5:     Let $h_t$ be the weak hypothesis returned by $\mathcal{W}$.
6:     Set loss: $\ell_t(p) = \sum_{i=1}^{m} p(i)(\frac{1}{\gamma} h_t(x_i) y_i - 1)$.
7:     Update: $p_{t+1} = \mathcal{A}(\ell_1, ..., \ell_t)$.
8: **end for**
9: **return** $\bar{h}(x) = \Pi\left(\frac{1}{\gamma T} \sum_{t=1}^{T} h_t(x)\right)$.

---

*Figure 4:* The algorithm has oracle access to either a $(\gamma, \epsilon_0, m_0)$-AWL algorithm (see Definition 6) or a $(\gamma, m_0)$-WL algorithm (see Definition 8). Both are denoted as $\mathcal{W}$. The optimizer is a $(\gamma, \mathcal{K}, T)$-OCO algorithm $\mathcal{A}$ (see Definition 1), where $\mathcal{K} = [0, 1]^m$ in the realizable case and $\mathcal{K} = [-1, 1]^m$ in the agnostic case. In line 4, we pass $(x_i, y_t^i)$ to $\mathcal{W}_i$, where $y_t^i$ is a random label s.t. $\mathbb{P}[y_i^t = y_i] = \frac{1+p_t(i)}{2}$. The final hypothesis "$\Pi\left(\frac{1}{\gamma T} \sum_{t=1}^{T} h_t(x)\right)$" is a randomized majority-vote, as defined in Equation 2.

## A.1   Solving Zero Sum Games Improperly Using an Approximate Optimization Oracle

Our framework uses as a main building block a procedure for approximately solving zero sum games using an approximate optimization oracle. It is described in this section.

In the zero sum games setting, there are two players A and B, and a payoff function $g$ that depends on the players' strategies. Player A's goal is to minimize the payoff, while player B's goal is to maximize it. Let $\mathcal{K}_A$ and $\mathcal{K}_B$ be the convex, compact decision sets of players A and B, respectively, and assume that $g$ is convex-concave. By Sion's minimax theorem [35], the value of the game is well-defined, and we denote it by $\lambda^*$:

$$\min_{p \in \mathcal{K}_A} \max_{q \in \mathcal{K}_B} g(p, q) = \max_{q \in \mathcal{K}_B} \min_{p \in \mathcal{K}_A} g(p, q) = \lambda^*$$

Let $\mathcal{K}_B'$ be a convex, compact set such that $\mathcal{K}_B \subseteq \mathcal{K}_B'$. We refer to strategies in $\mathcal{K}_B$ as proper strategies, while those in $\mathcal{K}_B'$ are improper strategies. We consider a modified zero sum games setting where the payoff function $g$ is defined on $\mathcal{K}_B'$, the set of improper strategies. Note that $\lambda^*$ is defined with respect to the set of proper strategies, and it is still a well-defined quantity in this game.

**Assumption 1:** Player B has access to a randomized approximate optimization oracle $\mathcal{W}$. Given any $p \in \mathcal{K}_A$, $\mathcal{W}$ outputs an improper best response: a strategy $q \in \mathcal{K}_B'$ such that $\mathbb{E}[g(p, q)] \geq \max_{q^* \in \mathcal{K}_B} g(p, q^*) - \epsilon_0$, where the expectation is taken over the randomness of $\mathcal{W}$.

**Assumption 2:** Player B is allowed to play strategies in $\mathcal{K}_B'$.

**Assumption 3:** Player A has access to a possibly randomized $(\mathcal{K}_A, T)$-OCO algorithm $\mathcal{A}$ with regret $R_{\mathcal{A}}(T)$ (See Definition 1).

**Algorithm 5** Improper Zero Sum Games with Oracles
---
1: **for** $t = 1, \ldots, T$ **do**
2:     Player A plays $p_t$.
3:     Player B plays $q_t \in \mathcal{K}'_B$, where $q_t = \mathcal{W}(p_t)$.
4:     Define loss: $\ell_t(p) = g(p, q_t)$
5:     Player A updates $p_{t+1} = \mathcal{A}(\ell_1, \ldots, \ell_t)$.
6: **end for**
---

**Proposition 12.** *If players A and B play according to Algorithm 5, then player B's average strategy $\bar{q} = \frac{1}{T}\sum_{t=1}^{T} q_t$, $\bar{q} \in \mathcal{K}'_B$, satisfies for any $p^* \in \mathcal{K}_A$,*

$$\lambda^* \leq \mathbb{E}[g(p^*, \bar{q})] + \frac{R_\mathcal{A}(T)}{T} + \epsilon_0,$$

*where the expectation is taken over the randomness of $\mathcal{W}$.*

*Proof.* Since the game is well-defined over $\mathcal{K}_A$ and $\mathcal{K}_B$, there exists a max-min strategy $q^* \in \mathcal{K}_B$ for player B such that for all $p \in \mathcal{K}_A$, $g(p, q^*) \geq \lambda^*$. Let $\bar{p} = \frac{1}{T}\sum_{t=1}^{T} p_t$, and observe that since the $p_t$'s depend on the sequence of $q_t$'s, they are also random variables, as well as $\bar{p}$. We have,

$$\mathbb{E}[\frac{1}{T}\sum_{t=1}^{T} g(p_t, q_t)] \geq \mathbb{E}[\frac{1}{T}\sum_{t=1}^{T} g(p_t, q^*)] - \epsilon_0 \geq \mathbb{E}[g(\bar{p}, q^*)] - \epsilon_0 \geq \lambda^* - \epsilon_0.$$

The first inequality is due to Assumption 1, where $\mathbb{E}[g(p_t, q_t)] \geq \max_{q \in \mathcal{K}_B} g(p_t, q) - \epsilon_0 \geq g(p_t, q^*) - \epsilon_0$. The second inequality holds because $g$ is convex in $p$.

Now, let $\bar{q} = \frac{1}{T}\sum_{t=1}^{T} q_t$; note that $\bar{q} \in \mathcal{K}'_B$ since $\mathcal{K}'_B$ is convex. For the upper bound, observe that the OCO regret guarantee implies that for any $p^* \in \mathcal{K}_A$ we have,

$$\mathbb{E}[\frac{1}{T}\sum_{t=1}^{T} g(p_t, q_t)] \leq \mathbb{E}[\frac{1}{T}\sum_{t=1}^{T} g(p^*, q_t)] + \frac{R_\mathcal{A}(T)}{T} \leq \mathbb{E}[g(p^*, \bar{q})] + \frac{R_\mathcal{A}(T)}{T},$$

where the second inequality holds because $g$ is concave in $q$. Combining the lower and upper bounds yields the theorem. $\square$

# B   Supplementary material for Section 4

**Lemma 13.** *Let $p^i, \mathcal{W}_i(x), y^i, y$ be random variables, such that $y, y^i \in \{\pm 1\}$, and $\mathbb{P}[y^i = y|p^i, y] = \frac{1+p^i}{2}$, $\mathbb{P}[y^i = -y|p^i, y] = \frac{1-p^i}{2}$. Moreover, $\mathcal{W}_i(x)$ and $y^i$ are conditionally independent given $p^i$ and $y$, namely $\mathbb{P}[\mathcal{W}_i(x), y^i|p^i, y] = \mathbb{P}[\mathcal{W}_i(x)|p^i, y]\mathbb{P}[y^i|p^i, y]$ Then $\mathbb{E}[\mathcal{W}_i(x) \cdot y^i] = \mathbb{E}[\mathcal{W}_i(x) \cdot yp^i]$.*

*Proof.*

$$\begin{aligned}
\mathbb{E}[\mathcal{W}_i(x) \cdot y^i] &= \mathbb{E}_{p^i,y}[\mathbb{E}[\mathcal{W}_i(x) \cdot y^i|p^i, y]] &\text{(law of total expectation)}\\
&= \mathbb{E}_{p^i,y}[\mathbb{E}[\mathcal{W}_i(x)|p^i, y] \cdot \mathbb{E}[y^i|p^i, y]] &\text{(conditional independence)}\\
&= \mathbb{E}_{p^i,y}[yp^i \cdot \mathbb{E}[\mathcal{W}_i(x)|p^i, y]] &(\mathbb{E}[y^i|p^i, y] = yp^i)\\
&= \mathbb{E}[\mathcal{W}_i(x) \cdot yp^i]
\end{aligned}$$

$\square$

### Proof of Theorem 11

We first state the following Lemma that will be used in the proof:

**Lemma 14.** *For any weak learner $(\gamma, T)$-WL $\mathcal{W}$, there exists $c = \tilde{O}(\sqrt{\sum_t p_t}) + 2R_\mathcal{W}(T)$ such that for any sequence $p_1, \ldots, p_T \in [0, 1]$,*

$$\sum_{t=1}^{T} p_t \cdot \mathcal{W}(x_t)y_t \geq \gamma \sum_{t=1}^{T} p_t - c.$$

*Proof.* The proof of this lemma is based on the proof of Lemma 1 in [9]. $\square$

We are now ready to prove Theorem 11. Let $h^*$ be an optimal hypothesis in hindsight for the given sequence of examples. We prove by lower and upper bounding the sum of losses. For simplicity of presentation we assume an oblivious adversary, however, using a standard reduction, our results can be generalized to an adaptive one [5]. Let $(x_1, y_1), ..., (x_T, y_T)$ be any sequence of observed examples. Observe that there are several sources of randomness at play; the weak learning algorithm $\mathcal{W}_i$'s internal randomness, the booster randomly passing the example to $W_i$ (line 5, Algorithm 3), and the randomized prediction (line 2, Algorithm 3). The analysis below is given in expectation with respect to all these random variables. We can now begin the analysis, starting with lower bounding the expected sum of losses, using the weak learning guarantee,

$$\frac{1}{\gamma}\mathbb{E}\Big[\sum_{i=1}^{N}\sum_{t=1}^{T}\mathcal{W}_i(x_t) \cdot y_t p_t^i\Big] \geq \mathbb{E}\Big[\frac{1}{\gamma}\sum_{i=1}^{N}\big(\gamma\sum_{t=1}^{T}p_t^i - \tilde{R}_{\mathcal{W}}(T)\big)\Big] \quad \text{(Weak learning (1, Lemma 14))}$$

$$\geq \sum_{i=1}^{N}\sum_{t=1}^{T}\mathbb{E}[p_t^i] - \frac{N}{\gamma}\tilde{R}_{\mathcal{W}}(T),$$

Thus, we obtain the lower bound on the expected sum of losses $\sum_t \sum_i \ell_t^i(p_t^i)$ (see Line 6 in Algorithm 1 for the definition of the $\ell_t^i$'s), given by,

$$\mathbb{E}[\sum_{t=1}^{T}\sum_{i=1}^{N}\ell_t^i(p_t^i)] \geq -\frac{N}{\gamma}\tilde{R}_{\mathcal{W}}(T).$$

For the upper bound, observe that the OCO regret guarantee implies that for any $t \in [T]$, and any $p_t^* \in [0, 1]$,

$$\mathbb{E}\Big[\frac{1}{N}\sum_{i=1}^{N}\ell_t^i(p_t^i)\Big] \leq p_t^*\left(\left(\frac{1}{\gamma N}\sum_{i=1}^{N}\mathbb{E}\big[\mathcal{W}_i(x_t)\big]\right)y_t - 1\right) + \frac{1}{N}R_{\mathcal{A}}(N),$$

Thus, by setting $p_t^*$ according to Lemma 5, and summing over $t \in [T]$, we get,

$$\mathbb{E}\Big[\frac{1}{N}\sum_{t=1}^{T}\sum_{i=1}^{N}\ell_t^i(p_t^i)\Big] \leq \sum_{t=1}^{T}(\mathbb{E}[\hat{y}_t]y_t - 1) + \frac{T}{N}R_{\mathcal{A}}(N).$$

By combining the lower and upper bounds for $\mathbb{E}\big[\frac{1}{NT}\sum_t\sum_i\ell_t^i(p_t^i)\big]$, we get,

$$\frac{1}{T}\sum_{t=1}^{T}\mathbb{E}[\hat{y}_t]y_t \geq 1 - \left(\frac{R_{\mathcal{W}}(T)}{\gamma T} + \frac{R_{\mathcal{A}}(N)}{N}\right).$$

## C  Supplementary material for Section 3

### C.1  Proof of Theorem 7

*Proof.* The proof has two parts. The first part is a straightforward reduction to the game-theoretic setup of Proposition 12, and the second part shows how to project the "improper" strategy obtained by Proposition 12 to the desired output hypothesis.

*Reduction to Proposition 12.* The agnostic version of Algorithm 4 can be presented as an instance of Algorithm 5, where Player A and B are the weak learner and the OCO oracle algorithms, respectively. The decision sets are $\mathcal{K}_A = [-1, 1]^m$, $\mathcal{K}_B = \Delta_{\mathcal{H}}$, and $\mathcal{K}_B' = \frac{1}{\gamma}\Delta_{\mathcal{H}}$, and the payoff function $g(\cdot, \cdot)$ is given by

$$g(p, q) = \sum_{i=1}^{m}p(i)(q(x_i)y_i - 1),$$

where $p \in \mathcal{K}_A$ is a vector in the $m$ dimensional continuous cube, and $q \in \mathcal{K}_B'$ is a non-negative combination of hypotheses in $\mathcal{H}$ (and so $q$ corresponds to the mapping $x \mapsto \sum_{h \in \mathcal{H}} q(h) \cdot h(x)$).

We leave it to the reader to verify that the agnostic weak learner corresponds to an approximate optimization oracle $\mathcal{W}$. Namely, for any $p \in \mathcal{K}_A$ the output $q' = \mathcal{W}(p)$ satisfies $q' \in \mathcal{K}'_B$ and

$$\mathbb{E}[g(p, q')] \geq \max_{q \in \mathcal{K}_B} g(p, q) - \frac{\epsilon_0 m}{\gamma}.$$

Furthermore, it can be shown that the value of the above game is

$$\lambda^* = m \cdot \max_{h \in \mathcal{H}} \operatorname{cor}_S(h) - m.$$

This can be done by (i) observing that the strategy $p = (1, 1, \ldots 1) \in \mathcal{K}_A$ is dominant for Player $A$ and (ii) computing $\max_{q \in \mathcal{K}_B} g(p, q)$ which is equal to $\lambda^*$ (since $p$ is dominating).

Now, Proposition 12 implies that for any $p \in [-1, 1]^m$, we have

$$m \cdot \max_{h \in \mathcal{H}} \operatorname{cor}_S(h) - m \leq \mathbb{E}\Big[\sum_{i=1}^m p(i)(\bar{q}(x_i)y_i - 1)\Big] + \frac{R_{\mathcal{A}}(T)}{T} + \frac{\epsilon_0 m}{\gamma}, \qquad (4)$$

where $\bar{q}(x_i) = \frac{1}{\gamma T} \sum_{t=1} h_t(x_i) \in \mathcal{K}'_B$.

*Projection.* Recall that the output hypothesis $\bar{h}$ is defined using the projection $\Pi$ (see Definition 2):

$$\bar{h}(x_i) = \Pi(\bar{q}(x_i)).$$

Now, by Lemma 5 there exists $p^*$ such that

$$m \cdot \max_{h \in \mathcal{H}} \operatorname{cor}_S(h) - m \leq \mathbb{E}\Big[\sum_{i=1}^m p^*(i)(\bar{q}(x_i)y_i - 1)\Big] + \frac{R_{\mathcal{A}}(T)}{T} + \frac{\epsilon_0 m}{\gamma} \qquad \text{(Equation 4)}$$

$$\leq m \cdot \mathbb{E}[\operatorname{cor}_S(\bar{h})] - m + \frac{R_{\mathcal{A}}(T)}{T} + \frac{\epsilon_0 m}{\gamma} \qquad \text{(Lemma 5)}$$

where the expectation is taken over the randomness of the projection, the weak learner, and the random samples given to the weak learner. Simple manipulation on the above inequality directly yields

$$\max_{h \in \mathcal{H}} \operatorname{cor}_S(h) \leq \mathbb{E}[\operatorname{cor}_S(\bar{h})] + \frac{R_{\mathcal{A}}(T)}{Tm} + \frac{\epsilon_0}{\gamma}.$$

If we use OGD as the OCO algorithm, we have $R_{\mathcal{A}}(T) = GD\sqrt{T}$, where $G \leq \frac{2\sqrt{m}}{\gamma}$ and $D = 2\sqrt{m}$. We arrive at the theorem by plugging in $\frac{R_{\mathcal{A}}(T)}{Tm}$. $\qquad\qquad\square$

## Proof of Theorem 9

*Reduction to Proposition 12.* Let $h^*$ be a concept consistent with the input sample (i.e. $h^*(x_i) = y_i$ for $i \leq m$) and let $\mathcal{H}' = \mathcal{H} \cup \{h^*\}$. It is convenient to define the decision sets are defined by $\mathcal{K}_A = [0, 1]^m$, $\mathcal{K}_B = \Delta_{\mathcal{H}'}$, and $\mathcal{K}'_B = \frac{1}{\gamma}\Delta_{\mathcal{H}'}$, and the payoff function $g(\cdot, \cdot)$ is again given by

$$g(p, q) = \sum_{i=1}^m p(i)(q(x_i)y_i - 1).$$

The weak learner corresponds to an approximate optimization oracle $\mathcal{W}$ with no additive error. That is, for any $p \in \mathcal{K}_A$ the output $q' = \mathcal{W}(p)$ satisfies $q' \in \mathcal{K}'_B$ and

$$\mathbb{E}[g(p, q')] \geq 0.$$

Next, one can show that the value of the game in this setting is $\lambda^* = 0$: indeed, this follows simce $\lambda^* = \min_{p \in \mathcal{K}_A} g(p, q^*) = 0$ and since the pure strategy supported on $h^*$, $q^* = q_{h^*} \in \mathcal{K}_B$ is dominant for player B. Applying Proposition 12, we have for any $p \in \mathcal{K}_A$, with $\bar{q}(x_i) = \frac{1}{\gamma T} \sum_{t=1} h_t(x_i) \in \mathcal{K}'_B$,

$$0 \leq \mathbb{E}\Big[\sum_{i=1}^m p(i)(\bar{q}(x_i)y_i - 1)\Big] + \frac{R_{\mathcal{A}}(T)}{T}. \qquad (5)$$

*Projection.* By the definition of $\bar{h}$, using Equation 5 and Lemma 5, we have

$$0 \leq \mathbb{E}[\operatorname{cor}_S(\bar{h})] - 1 + \frac{R_{\mathcal{A}}(T)}{Tm}.$$

As before, using OGD as the OCO algorithm $\mathcal{A}$ yields $\frac{R_{\mathcal{A}}(T)}{Tm} = O(\frac{1}{\gamma\sqrt{T}})$.

## Footnotes

[4]Note that when $p_t = \mathbf{0}$ is constantly zero then the distribution used in the realizable setting is not well defined. There are several ways to circumvent it. Concretely, we proceed in such case by setting $h_t = h_{t-1}$ and proceeding to step 6.

[5] See discussion in [12], Pg. 69, as well as Exercise 4.1 formulating the reduction.