[Reviews · NeurIPS 2020]

Review 1

Summary and Contributions: In this work, the authors propose a Boosting algorithm applicable to the online agnostic case. The primary motivation to do so is that the assumption in the realizable setting, contrary to the agnostic setting, that there is a hypothesis with a near-zero error that does not hold in real-world scenarios. This paper's main contribution is frameworks, which transform boosting into online convex optimization (OCO) that is used to perform the relabeling/reweighting. The authors propose such a framework for Online Agnostinc Boosting with OCO, Online Realizable Boosting with OCO, and (Statistical) Boosting with OCO. The first two are the main focus of the paper, while the latter is further analyzed in the appendix.

Strengths: The paper is well written. The stated claims are supported by proofs, and the simplifications made seem sound.

Weaknesses: The related work section of this paper is quite short and only focuses on boosting. An overview of online convex optimization would be helpful, as it is a central part of the proposed frameworks. This paper's motivation is that the assumption for realizable boosting, mentioned above, is not standard for online learning. However, there is no comparison against state-of-the-art bounds or an analysis, which shows in which cases the regret bounds are superior, i.e., more realistic, than the bounds of the state-of-the-art approaches.

Correctness: Seems to be the case.

Clarity: The paper is well written and easy to read. The investigated scenarios are compared to each other, and differences are explained. However, as someone not familiar with this topic, some terms, such as lambda, R_W(T), and R_A(T), seem opaque to me, as it is not clear to me what they represent.

Relation to Prior Work: The prior work is discussed, and the gap between the state-of-the-art and this work is clearly shown.

Reproducibility: Yes

Additional Feedback: none


Review 2

Summary and Contributions: This paper introduces an agnostic online learning boosting algorithm for classification, extending prior work on boosting in the realizable case (both in the statistical and online cases) and on agnostic boosting in the statistical learning setting. Given access to "agnostic weak online learners" (satisfying some weak error bound), the procedure constructs an online learner with a regret bound. This relies on a generic reduction of boosting to online convex optimization, which is applied to other settings: online-realizable, statistical-agnostic and statistical-realizable, albeit the results in the realizable case are worse than the known optimal bounds.

Strengths: This contribution fills a gap in the literature, by providing boosting algorithms in the agnostic online setting, complementing previous results in the statistical setting or in the realizable case. I found this interesting, although I am not familiar enough with the boosting literature to confidently assess the significance of this contribution (hence my low confidence score). The specific reductions to OCO provided here may also be of interest, at least in the agnostic setting.

Weaknesses: The specific reduction to OCO, which is a main contribution of this work as it can be applied to several settings, leads to suboptimal bounds when applied to realizable boosting. Also, the interest of the ability to plug-in a generic OCO procedure (which is presented as a contribution) is unclear, since the OCO problems considered in the online setting in the main paper are actually two-experts aggregation problems, for which very simple algorithms (OGD, Exponential Weights) achieve minimax regret. Finally, the adopted boosting setting is not justified. One approach to boosting (see, e.g., "On the Rate of Convergence of Regularized Boosting Classifiers", G. Blanchard, G. Lugosi, N. Vayatis, J. Mach. Learn. Res., 2003) is to construct, given a base class of simple predictors, an aggregated rule comparing favorably to linear/additive combinations of such predictors. Here the adopted definition (in line with previous work on agnostic boosting) is different, with a fixed class of comparison predictors, and as weak learner some algorithm satisfying a weak relative error bound, and the objective being to obtain some regret bound. It would be helpful to motivate this definition, by providing at least one specific example (half-spaces? decisions trees?) where such a problem might arise, indicating the class H and weak learners.

Correctness: Overall, the work appears to be theoretically sound, as I did not see any mistake in the portions I checked. I did not get the main idea behind the provided OCO reduction, though, as the exposition is a bit terse in these sections.

Clarity: I found the paper to be somewhat lacking in terms of clarity, and oftentimes hard to follow. Some of it may simply be due to the fact that I am not an expert on boosting, but there were also issues in the basic definitions (see additional remarks). Overall, the current version feels somewhat unpolished and terse at times, and a better exposition could help with readability. (This is a main issue I have with this submission.)

Relation to Prior Work: The work does refer to previous work on the topic, and discusses how it differs from it. I would suggest discussing the previous work on online boosting in the statistical setting more in detail (including in the "related work" section), and discussing differences between these approaches and the reduction adopted here.

Reproducibility: Yes

Additional Feedback: * Definition 1 is unclear: - In the current definition, an AWOL is characterized by (gamma,T), while R_W(T) is left implicit. As the definition is written, it seems to mean that W is an AWOL if there exists some R_W such that the regret inequality is satisfied. However, any algorithm is an AWOL for a sufficiently large R_W (see the point below). Hence, R = R_W (T) should be part of the definition. - The assumption that the function R_W is sublinear is immaterial, since the definition holds for a fixed horizon T (sublinearity is an asymptotic property of a sequence). In particular, for any fixed T, one can take R_W (t) = 4T, and then any algorithm is a (gamma,T)-AWOL. One option to fix this would be to remove T from the definition, and say that a *sequence* of algorithms indexed by T is an AWOL if there exists a sublinear function R_W such that for every T, the bound holds (one could also consider a single, anytime algorithm, instead of a sequence of algorithms). Another one would be to keep the fixed sample size, and add R = R_W(T) to the definition (note that for a fixed sample size T, only the value of R_W at T matters, hence R_W should be replaced by the single number R_W (T)). - The usual definition of regret only corresponds to the case gamma=1 in the definition. (Also, although this is not written in the definition, gamma is presumably assumed to be in (0, 1], with the case gamma<1 corresponding to "weak" learners.) A bound of the kind of Lemma 1 corresponds to some relative loss bound, or simply "error bound". * It would be worth providing the main idea behind the proposed reduction (and, ideally, how it differs from previous OCO reductions), as opposed to only a derivation/proof. This could help with readability. * The setting could ideally be illustrated by providing at least one specific example (see remark above in "limitations"). * Can't the results in the online setting by directly transferred to the statistical case (excess risk bounds) through online-to-batch conversion, without different reductions? (The sample complexity provided in the statistical case are the same as that of the online case, after all.) This seems possible at least as long as the base learners are AWOL as opposed to AWL. It may be worth mentioning this, and discussing the relations and differences between AWOL and AWL. * Theorem 2: it is worth recalling the definition of regret, which is not provided at this point (and which should be distinguished from the weaker inequality of Lemma 1). * Line 78: "a regret of epsilon is obtained": shouldn't it be epsilon*T instead? * It may be worth mentioning that the AWOL definition is an online analogue of the agnostic weak learning assumption in the statistical setting by Kalai and Kanade. * The paper mentions a reduction to Online Convex Optimization (OCO), yet in the algorithms (at least those in the main text about online boosting) only the specific case of prediction with two experts' advice is used (that is, linear loss functions on a segment), as opposed to general OCO. Introducing (and referring to) this explicit setting could simplify the presentation (by removing the unneeded generality and reference to a generic K). (A different domain is used in the statistical setting in the appendices, though.) * The presentation of the algorithm in Section 2 is hard to follow. For instance, it could be helpful to indicate the explicit time dependence of the W_i, and replace the OCO setting with general domain K with the expert setting. * Line 97: shouldn't "statistical" be "realizable" instead? == Update in light of the authors' response: I would like to thank the authors for their response. After reading it and the other reviews, I still have concerns with the clarity, which I feel should be addressed before publication. (Aside from those, I do not have strong recommendations on the decision.) - On the OCO formulation: it is unclear what one should expect to gain from the formulation in terms of general OCO of the simple two-expert problem, or what insight it brings in this context. In terms of clarity, I found this to make reading slightly harder by hiding the nature of the points/domain. - On the definition (and dependence on horizon T vs. function R_W): as noted by the authors, AWOL is thus a property of a sequence of algorithms. This conflicts with the (gamma,T) definition, where a fixed horizon T is part of the definition, while the bound R_W is not (it should probably be the opposite). Note also (as noted above in the review) that R_W is referred to in Def 1 as the regret, while the bound in Def 1 is not a regret bound due to the presence of gamma; while Thm 2 refers to a (standard) regret bound without a definition in-between. More rigor would help at this stage, since this is the basic definition of the problem setting. - Finally, the overall clarity of the exposition could be improved. For instance, it would be helpful to convey the main ideas, specify the setting when possible (see above), and indicate differences in approaches and difficulties when extending previous results in the statistical case to the online setting.


Review 3

Summary and Contributions: The authors provide an agnostic online boosting algorithm that given a weak learner boosts it to a strong learner with sublinear regret. The algorithm works based on a reduction to online convex optimization, which efficiently converts an arbitrary online convex optimizer to a boosting algorithm.

Strengths: The problem is motivated and interesting, and is relevant to the NeurIPS community. The claims are baked up with theoretical guarantees, and comparison with state-of-the-art guarantees is discussed.

Weaknesses: The main drawback of the paper is that there is no empirical evaluation to show the effectiveness of the proposed method on real online-learning tasks. In particular, since an important contribution of this work is reducing boosting to online convex optimization. For settings that were previously studied, performance comparison with state-of-the-art method would significantly strengthen the work. Although, authors mention the inferior guarantees of their proposed general framework compared to [34, [9], [28], it'd be still very interesting to see how much one may loose by using the proposed framework compared to the state-of-the-art less general techniques.

Correctness: The claims seems to be technically correct, however no empirical evaluation is provided.

Clarity: The paper is well-written, but would significantly benefit from restructuring, by moving some of the proofs to the appendix and adding experimental evaluations. Also, it'd help if the authors can clarify some of the notations, e.g., what is p in line 6 of algorithm 1 and how is it different than p_t^i?

Relation to Prior Work: The relations to prior work and the discussion of how this work differs from previous contributions is provided.

Reproducibility: Yes

Additional Feedback: --- update --- I thank the authors for their response. I understand that the main contribution of the paper is the theoretical results. While I appreciate the theoretical contributions, I still believe that adding some experiments to show the performance of the proposed method in scenarios where existing methods do not provide any theoretical guarantee would help making the contribution more concrete to the readers.

[Author Response · NeurIPS 2020]

We thank the reviewers for reading the manuscript and for their feedback and suggestions. We will correct typos pointed out by the reviewers.

**Reviewer 1**   We will add an overview of Online Convex Optimization to provide helpful background, as the reviewer suggested. Regarding the agnostic setting being more realistic; we first note that previous work on realizable agnostic boosting operates under the assumption that an access to a realizable weak learning oracle is given, meaning that the number of mistakes made by the weak learner is bounded, and the set of allowable sequences is restricted. In most real-world problems this assumption does not hold. In the agnostic setting that assumption is removed, and the learner is only assumed to compete with the best predictor in a given class, where the input sequence is not restricted and can be chosen adversarially.

**Reviewer 3**   Online agnostic boosting is the primary contribution of this work, and we provide the first algorithm in this setting (which therefore cannot be compared to previous works). The application of our methodology to the other 3 settings is a secondary contribution, which serves as a testament to the generality of the OCO framework. In particular we did not intend to improve upon the state of the art here. The goal of introducing results in the other 3 settings was to demonstrate the generality and uniformity of our approach. Indeed, our approach does not achieve quantitative improvement in the 3 settings that were previously studied, but merely demonstrates an abstraction that applies to all 4 cases.

- "interest of the ability to plug-in a generic OCO": First, we believe that an abstract reduction to a generic OCO (rather than e.g., specifically using OGD) yields a more modular proof which is simpler to comprehend (and a more general statement). In addition, one can imagine specific constrained settings in which specifically tailored OCOs will exhibit superior performance compared to general-purpose algorithms such as OGD/Exponential-Weights/etc.

- "the adopted boosting setting is not justified": Indeed our definitions are an immediate adaptation of the work [28] (by Kanade and Kalai) to the online setting. We remark that [28] is the most recent work on agnostic boosting, in a well-studied learning setting, and thus it is justified in the literature. As an example for such a problem, consider a weak online learner for a class of experts, and assume that the data comes from a majority vote over $N/2$ experts. Notice that learning the class of majority votes over $N/2$ experts is exponential in $N$, whereas our framework enables a mechanism efficient in $N$ for learning that class. We will elaborate on such examples in the final version of the paper.

- "The assumption that the function $R_W$ is sublinear is immaterial": The underlying assumption is that there is a sequence of algorithms, one per each horizon-size $T$, with a single sublinear function $R_W(\cdot)$. This is in fact an assumption commonly used in online learning (see e.g., [23]). We need $R_W(T)/T$ to converge to zero, and hence the average regret will converge to zero. If we use an algorithm without this property, then the regret bound remains a large constant, and can be vacuous.

- "online-to-batch": Indeed, online-to-batch enables us to convert a weak online learner into a strong statistical learner. However this is a weaker result than converting a weak statistical learner to a strong statistical learner (which is what boosting algorithms in the statistical settings achieve). That is, the online weak learning assumption implies the statistical weak learning assumption, but not vice versa.

- "a regret of $\epsilon$ is obtained": this is a typo, and should be "an average regret of $\epsilon$ is obtained".

- We agree with the reviewer's suggestions on having a better exposition to improve readability for readers less familiar with the boosting literature, and will incorporate that in our revision.

**Reviewer 4**   First, we note that the focus of this paper is the theoretical investigation of online boosting reductions to regret minimization. Let us stress that the main contribution, i.e. the derivation of an **agnostic** online boosting algorithm, answers a basic question in a well-studied learning setting, and there are no previous results in that setting to compare against. Nevertheless, it might indeed be interesting to experimentally compare our algorithm to the known online boosting algorithms in the **realizable** setting.

- Regarding notation, $p$ in line 6 of Algorithm 1 is a parameter of the function (for any $p \in [-1, 1]$), and $p_t^i \in [-1, 1]$ is a value that is defined by the algorithm (as in line 4), and is essentially the output of the OCO algorithm $\mathcal{A}$ after observing all losses up to $t, i - 1$. We will try to further clarify this in the paper.

[Meta-Review · NeurIPS 2020]

Online boosting has seen a lot of activity recently. This excellent contribution extends the theory to non-realizable case which is also the more practical case.